# High acceptability of newborn screening for sickle cell disease among post-natal mothers in Western Kenya

**John Orimbo**[1]*, **Shehu Shagari Awandu**[1], **Faith Muhonja**[2], **Patrick Owili**[3], **Dickens Omondi**[1]

**1** Department of Biomedical and Nursing Sciences, School of Health Sciences, Jaramogi Oginga Odinga University of Science and Technology, Bondo, Kenya, **2** Department of Public Health, Amref International University, Nairobi, Kenya, **3** African Population and Health Research Centre, Nairobi, Kenya

* jguda006@gmail.com

## Abstract

Sickle cell disease is a genetically inherited blood disorder that manifests early in life with resultant significant health complications. Globally, nearly three quarters of all affected babies are in sub-Saharan Africa. Early identification of babies with sickle cell disease through newborn screening followed by early linkage to care is recommended. However, the program has not been widely adopted in the sub-Saharan Africa. The Kenyan ministry of health, in 2020, published a policy on newborn screening for sickle cell disease from levels 2–6 healthcare facilities. However, evidence on acceptability of newborn screening to scale up newborn screening program is scarce. Few studies have been conducted across the sub-Saharan Africa to assess the acceptability of newborn screening for sickle cell disease with conflicting results. This study assessed factors associated with acceptability of newborn screening among mothers of newborns delivered at Homa bay county teaching and referral hospital, western Kenya. This study employed a cross-sectional design among postnatal mothers at Homa bay county teaching and referral hospital with 399 postnatal mothers enrolled into the study. After obtaining informed consent from the postnatal mothers, a semi-structured questionnaire was used for data collection. Maternal sociodemographic characteristics, knowledge, and perception were assessed. Babies were also screened for sickle cell disease using Sickle SCAN point-of-care test. The acceptability was calculated as percentage of mothers accepting to have their babies screened. Data were analyzed using logistic regression to explore factors associated with acceptability of newborn screening for sickle cell disease. Ninety-four percent of mothers accepted newborn screening for sickle cell disease. Only maternal age and occupation were significantly associated with acceptability of newborn screening for sickle cell disease. Mothers aged 25−34 years were 3 times less likely to accept newborn screening for sickle cell disease than those younger mothers than 25 years (OR=0.33; 95% CI = 0.13–0.86; $p$ = 0.024). Similarly, mothers in the formal employment

**Data availability statement:** All relevant data are within the paper and its Supporting Information files.

**Funding:** JO,Grant Number CSA2020E-3139-CDAE,Funded by European and Developing Countries and Clinical Trials Partnership (EDCTP) program of the European Union. Sponsors did not play any role in study design, data collection and analysis ,preparation of the manuscript or decision to publish.

**Competing interests:** The authors have declared that no competing interests exist.

**Abbreviations:** aOR, Adjusted Odds Ratio; CI, Confidence Interval; EC, Ethics Committee; JOOUST, Jaramogi Oginga Odinga Teaching University of Science and Technology; NACOSTI, National Commission Of Science, Technology and Innovation; NBS, Newborn Screening; OR, Odds Ratio; SCD, Sickle Cell Disease; SCT, Sickle Cell trait; SSA, Sub-Saharan Africa.

were 6 times less likely to accept newborn screening for sickle cell disease than those who were students (OR= 0.16; 95%CI = 0.03–0.84; $p = 0.031$).Mothers who were in formal employment were 25 times less likely to accept newborn screening for sickle cell disease than those who were students in the multivariate logistic analysis model (aOR= 0.04; 95% CI = 0.00–0.78$; p = 0.034$). The acceptability of newborn screening for sickle cell disease is high in the county. The Homabay county ministry of health should implement routine newborn screening for sickle cell disease in all healthcare facilities conducting deliveries of newborns.

## Introduction

Sickle cell disease (SCD) is a major genetically inherited blood disorder, caused by autosomal recessive inheritance, that manifests early in life with resultant significant health complications. Globally, nearly three quarters of all sickle cell disease-affected babies are in sub-Saharan Africa [1]. In 2011, the World Health Organization (WHO) recommended an increased awareness, improved access to health services, and provision of technical support for prevention and management of sickle cell disease [2]. Programs such as newborn screening for sickle cell disease have shown great success towards preventing and managing the disease [3]. However, only a few countries in Africa have piloted the screening program and none has scaled up the implementation [4–6]. Successful implementation of newborn screening for sickle cell disease followed by early initiation of penicillin prophylaxis, pneumococcal vaccines, and hydroxyurea treatment are likely to reduce the disease burden in low- and middle-income countries (LMICs).

Nonetheless, limited data on the prevalence of sickle cell disease at birth in Africa makes it difficult to quantify the magnitude of the disease burden. Additionally, approximately 90% of the children with sickle cell disease die before their fifth birthday, partly because of late diagnosis [7]. This can be solved through adequate implementation of newborn screening for sickle cell disease. This implementation however, heavily depends on its acceptability [8] and socio-cultural settings [9], which vary from one community or region to another. Unacceptability of the screening program for sickle cell disease by the parents or caregivers or authorities in Africa can greatly hinder effective implementation of the program [4]. There is need to assess the acceptability of the program before its implementation which may not only promote public participation but also community ownership of the intervention.

The prevalence of sickle cell disease varies on the malaria endemicity patterns. The disease burden in Kenya is quite high especially around the lake regions, in western as well as the Coastal regions [10]. The prevalence of sickle cell disease around western Kenya at birth is about 4.5% for sickle cell disease, and 18% for sickle cell trait [11].

Homa bay county is yet to implement newborn screening for sickle cell disease. This is despite the high prevalence of sickle cell disease among infants who were accessing maternal and child health services in the county, which was found to be at

9.6% [12]. The Kenyan ministry of health, in 2020, published a policy on newborn screening for sickle cell disease from levels 2–6 healthcare facilities in areas of high sickle cell disease prevalence, one of which is Homa bay county. Routine newborn screening for sickle cell disease is recommended in regions where sickle cell disease prevalence at birth is 0.05% or more [13]. There is however, paucity of data on the acceptability of newborn screening in the county. Acceptability is important for successful implementation of the intervention program [8]. Additionally, extrapolating data from another county may not be feasible owing to the variable nature of acceptability [14].

Here, we assessed the acceptability of newborn screening for sickle cell disease and its associated factors in Homa bay county, western Kenya. This is important in promoting the establishment of a systematic newborn screening program for sickle cell disease in Homa bay county, and subsequently a successful implementation of the intervention [8]. The results of this study are critical to the ministry of health at the county-level for planning of the sickle cell disease screening program in the county.

## Methods

### Study design, study population and study setting

This study employed a cross-sectional design at the postnatal wards in the Homa bay county teaching and referral hospital, Homa bay county, western Kenya. Homa bay county teaching and referral hospital is the county referral hospital with a wider catchment across the eight sub counties in Homa Bay County. The study involved postnatal mothers in the postnatal wards between 14th April 2023 and 15th June 2023. The site has a sickle cell disease clinic, however, it lacks newborn screening for sickle cell disease implementation.

### Sample size and sampling technique

Cochran's formula for calculating the sample size when the population is infinite was used. A maximum variability of 50% was used to estimate the maximum sample size required since the acceptability of newborn screening for sickle cell disease was unknown in Homa bay county. The final sample size was 403 mothers, after adjusting for 5% non-response. Consecutive sampling technique, where all participants meeting eligibility criteria were enrolled until required sample is achieved, was used to select the participants. All postnatal mothers available at the postnatal ward during the study period, and willing to take part in our study and met the inclusion criteria were selected and enrolled. The inclusion criteria were: The postnatal mother giving an informed consent or assent to participate in the study and whose outcome of delivery was an alive infant. Postnatal mothers who were medically unstable to respond to the questionnaire or needed urgent critical care were excluded from participating in the study.

### Data collection tools

Kobo collect tool was used to collect the data using a semi-structured questionnaire with five sub-sections. The first section collected demographic information: Maternal age, residence, marital status, occupation, religion, education level and parity. The second section contained questions on the knowledge (awareness) of the mother about sickle cell disease including ever having heard of or being aware of sickle cell disease, being aware of their haemoglobin status, knowledge about sickle cell disease being a blood disorder, knowledge on transmission of sickle cell disease, when it can be diagnosed and knowledge on the existence of newborn screening for sickle cell disease programs. The third section contained the questions on the perceptions of the newly delivered mothers. The perception was measured by asking question on their opinion on to when screening for sickle cell disease should be done, ethical issues and the perceived effectiveness of the newborn screening for sickle cell disease. The perception was considered good when in line with the national government newborn screening policy. The government policy launched by the Kenya ministry of health in 2020 advocated for newborn screening within the first 6 weeks of birth as an effective tool to control sickle cell disease. The fourth section

of the questionnaire explored the possible intervening variables mainly, the cultural issues around newborn screening and the role of the male partner in making the decision as to whether to accept newborn screening for sickle cell disease. The fifth section collected data on their intent/willingness to have their newborns screened and those who were willing had their newborns screened using sickle SCAN kit (BioMedomics Inc, NC, USA). The sickle SCAN test kit is capable of identifying Hemoglobin A, S and C variants in blood samples.

## Ethical approval and consent to participate

The proposal was approved by the board of postgraduate studies of Jaramogi Oginga Odinga University of Science and Technology. Ethical approval for the study was obtained by JOOUST Ethics Committee (ERC 36/02/23–33) and the NACOSTI permit obtained from the National Commission of Science, Technology and Innovation (Research License 101836). Using the approvals from the Ethic Committee (EC) and NACOSTI, the County authorization was obtained from the county health management team and the county director of medical services and notice issued to the hospital management and the postnatal department. The research assistants signed a confidentiality agreement form before starting the data collection and were adequately trained to minimize any potential harm to the participants. Informed consent was obtained from the study participants' mothers prior to any study procedure. The research assistants took the participants through the informed consent forms for adult mothers and informed assent forms for the minor mothers in the participants language of choice. Informed consent documents were translated from English to Kiswahili and Luo. language. The data collected was kept confidential and, in a password-protected computer. Data was entered directly into the password-protected computer with limited access. The access of the research assistants was terminated immediately data cleaning was complete. No paper data was collected in the study. Key personal identifying information such as name of the participant were not captured in order to deidentify the participant and minimize chances of harm from breach of confidentiality. Mothers of newborns who were willing to have their newborns screened were offered psychological counselling pre and post the test. Mothers whose newborns screened positive and need advanced psychological counseling were referred to the counselling department of the Homa bay county teaching and referral hospital. The newborns who screened positive for sickle cell disease were linked with the sickle cell disease clinic of the Homa bay county teaching and referral hospital for confirmatory testing and subsequent early initiation of comprehensive care, usually initiated at the age of 2 months. The mothers of the newborns who were found to have sickle cell traits were offered genetic counselling by the study nurses. The risk to the newborns screened for sickle cell disease included minimal pain, swelling and bruise at the sample collection site due to the prick but this was minimized by having the blood sample collected by adequately trained and professional research assistants with experience in neonatal blood sampling from the heels. The heel prick was done while maintaining the infection prevention control measures like adequate sterilization of the prick site to minimize chances of prick site infection.

## Enrollment and data collection

A total of 399 participants were enrolled into the study, representing 99% response rate. Written informed consent was obtained from potential participants' mothers who were 18 years and above. A written informed assent was obtained from minor mothers (those less than 18 years and lacking legal capacity to consent on their own) in addition to a written consent from their parents or guardians. The potential participants were taken through the consenting process by a research assistant on a one-by-one basis, explaining the elements of the informed consent document. The informed consents and assents had adequate details on sickle cell disease and their participation in the study, and also on risk and benefits of participation. This was made so in order to provide the potential participants with adequate prior information for making an informed decision before signing the informed consent or assent (for minor mothers). The participants were provided with a copy of the signed consent or assent form (for the minor mothers) and another copy retained by the investigator.

 

After consenting, a semi-structured questionnaire was administered to each participant by the research assistants. The research assistants comprised of 1 clinical officer/physician assistant, 2 nurses, 1 laboratory technician and 2 counsellors. The principal investigator trained the research assistants prior to the start of the study and supervised data collection. The mothers were finally asked about their intent/willingness to have their newborns screened. Willing participants had their babies screened for sickle cell disease using the Sickle SCAN® point-of-care test following manufacturer's instructions (BioMedomics Inc.,NC, United States). The results were interpreted as sickle cell trait if with 3 lines visible: at the control, HbA and HbS; sickle cell disease when two lines appeared, at the control and HbS or no haemoglobinopathy when two lines appeared, at the control and HbA. The laboratory technician conducted the screening test at the bedside following the manufacturer's instructions. The newborns' heels were pricked using 1.2 mm lancet pricker, and 2 drops of blood collected directly into the device. The results were ready in 3–5 minutes. The study nurses offered psychological counselling to all the mothers pretest and to mothers whose babies had sickle cell trait or sickle cell disease post-test. The mothers whose newborns screened positive for sickle cell disease were referred to the sickle cell disease comprehensive clinic and also received psychological counselling from the study team.

### Statistical analyses

The data were entered into excel for cleaning before analysis using STATA version 16.0 (StataCorp, 2019). The level of acceptability of newborn screening for sickle cell disease was analyzed using descriptive statistics only. Even though acceptability is a multidimensional construct, in this study it was defined as the willingness of the mother to accept newborn screening for sickle cell disease. The percentage acceptability, calculated as the percentage of mothers who agreed to have their newborns screened for sickle cell disease.

The factors associated with acceptability of newborn screening for sickle cell disease were analyzed using both descriptive statistics and inferential statistics. Descriptively, proportions of acceptability among various factors were calculated and represented in a table. Inferentially, Chi-square and Fisher's exact test (where at least a cell has a count less than 5) were used to assess the association between the sociodemographic, maternal knowledge and maternal perception characteristics and the acceptability of newborn screening for sickle cell disease. The factors that had statistically significant association with acceptability were further analyzed using logistic regression analysis to establish the point estimate and the magnitude of association. After the unadjusted analysis, statistically significant variables, were fit into a multivariate logistics regressions model using forward stepwise regression approach. Both unadjusted and adjusted odds ratio (OR) and 95% confidence intervals (95% CI) were reported. The results of the newborn screening were used to calculate the prevalence of sickle cell disease and sickle cell trait among the newborns.

## Results

### Summary of participant characteristics

The mean age of the mothers was 26.2 years (SD 6.07) with 25–34 years being modal age-group, comprising 44.1% of the respondents. Majority (300/399; 75%) of respondents were married; 59.7% (238/399) had one or 2 children; 41.4% (165/399) had attained secondary level education while, 70% were engaged in either informal or no employment. Only 2 mothers (0.5%) reported to have had a previous child with Sickle cell disease. Majority (371/399; 93%) of the respondents had ever heard about sickle cell disease; only 4% knew their sickle cell disease status with only 2% having ever been screened for sickle cell disease before; over 90%(362/399) knew the cause of sickle cell disease; only 7%(28/399) of respondents believed that sickle cell disease was as a result of God's will with less than 2%(5/399) believing sickle cell disease was due to God's punishment or evil spirits and majority (386/399;96.7%) believed the best timing for sickle cell disease testing was during the postnatal period.More than 50% of the postnatal mothers were from Homa Bay town subcounty (Table 1).

**Table 1. Sociodemographic characteristics of the postnatal mothers N = 399.**

| Characteristics | Frequency | Percentage |
|---|---|---|
| Age group | | |
| 11-24 years | 175 | 43.9 |
| 25-34 years | 176 | 44.1 |
| >35 years | 48 | 12.0 |
| Education level | | |
| Tertiary | 120 | 30.1 |
| Secondary | 165 | 41.4 |
| Primary | 112 | 28.1 |
| No Education | 2 | 0.5 |
| Marital Status | | |
| Married | 300 | 75.2 |
| Widowed | 5 | 1.3 |
| Separated | 21 | 5.3 |
| Never married | 73 | 18.3 |
| Occupation | | |
| Student | 72 | 18.1 |
| Formal employment | 40 | 10.0 |
| Informal employment | 226 | 56.6 |
| No employment | 61 | 15.3 |
| Religion | | |
| Roman catholic | 73 | 18.3 |
| Protestant | 206 | 51.6 |
| Pentecostal | 55 | 13.8 |
| Muslim | 6 | 1.5 |
| Others | 59 | 14.8 |
| Parity | | |
| One child | 141 | 35.3 |
| Two children | 97 | 24.3 |
| Three children | 80 | 20.1 |
| Four children and above | 81 | 20.3 |
| Previous child with sickle cell disease | | |
| Yes | 2 | 0.5 |
| No | 397 | 99.5 |
| Death of under 2 years | | |
| Yes | 29 | 7.3 |
| No | 370 | 92.7 |
| Subcounty of residence | | |
| Homa Bay town | 206 | 51.6 |
| Rachuonyo North | 30 | 7.5 |
| Rangwe | 38 | 9.5 |
| Rachuonyo East | 28 | 7.0 |
| Rachuonyo South | 25 | 6.3 |
| Suba South | 22 | 5.5 |
| Suba North | 21 | 5.3 |
| Ndhiwa | 29 | 7.3 |

### Level of acceptability of newborn screening for sickle cell disease among the postnatal mothers

Out of the total 399 mothers interviewed, 375/399 (94.0%) agreed to have their newborns screened for sickle cell disease using the sickle scan point of care kit. The level of acceptability was therefore 94.0%.

### Association between sociodemographic characteristics and acceptability of newborn screening for Sickle cell disease among the postnatal mothers

Chi-square method was used to assess association between the sociodemographic characteristics of the respondents and their acceptability of newborn screening for sickle cell disease. Maternal occupation *(p = 0.026)* and maternal age *(p = 0.023)* were the only factors that had statistically significant association with the acceptability of newborn screening for sickle cell disease. Respondents who were students and those younger than 25 years were more likely to accept newborn screening for sickle cell disease. The student as an occupation category comprised mothers who were only school going and therefore not in any other occupation. Characteristically in this study, about 90% (65/72) were younger than 25 years. Majority of the students 91% (66/72) had post primary education, with the modal group in secondary schools. The odds of accepting newborn screening for sickle cell disease among mothers who were in formal employment was lower than those who were student (OR=0.16; 95%CI = 0.03–0.84; *p* =0.031). Mothers aged between 25−34 years of age had lower odds of accepting newborn screening for sickle cell disease than those younger than 25 years (OR=0.33;95%CI = 0.13–0.86; *p* =0.024). No significant association was found between acceptability of newborn screening for sickle cell disease and other sociodemographic characteristics. (Table 2).

### Association between maternal knowledge and acceptability of newborn screening for sickle cell disease among the postnatal mothers

Fisher's exact method was used to assess the association between maternal knowledge, measured by finding what they know about sickle cell disease as shown in Table 3 below, and the acceptability of newborn screening for sickle cell disease, since the criteria for chi-square was not met. Most of the cells had values less than 5. No statistically significant association was found between maternal knowledge dimensions and acceptability of newborn screening for sickle cell disease. Most of the mothers had ever heard about sickle cell disease and knew it is a non-contagious blood disorder. However, only few mothers had ever been screened for sickle cell disease nor knew their sickle cell disease status. Most had good knowledge of the inheritance patterns of sickle cell disease and nearly two thirds had never heard about new-born screening for sickle cell disease before (Table 3).

### Association between acceptability of newborn screening for sickle cell disease and perception of the postnatal mothers

Fisher's exact method was used to assess the association between maternal perception and the acceptability of newborn screening for sickle cell disease, since the criteria for chi-square test was not met. Most of the cells had values less than 5. No statistically significant association was found between maternal perceptions and acceptability of newborn screening for sickle cell disease. Postnatal mothers however had good perception about sickle cell disease. Most of the postnatal mothers had good perception of what causes sickle cell disease, with most believing that the best time to screen for sickle cell disease is during the newborn period (0–6 weeks). Majority of the postnatal mothers believed newborn screening for sickle cell disease was morally right and would be useful in controlling the disease (Table 4).

### Multivariate logistic regression analysis of factors with significant association with acceptability of newborn screening for sickle cell disease at bivariate analysis

Mothers in the formal employment were 25 times less likely to accept newborn screening for sickle cell disease than those who were students (aOR= 0.04; 95% CI = 0.00–0.78*; p* =0.033) at multivariate logistic regression analysis. Male partner

**Table 2. Association between acceptability of newborn screening for sickle cell disease and Sociodemographic characteristics of the postnatal mothers.**

| Characteristic | NBS YES | NBS NO | X² value | df | p-value |
|---|---|---|---|---|---|
| Marital Status* | | | | 3 | 0.181 |
| Married | 279(93.00) | 21(7.00) | | | |
| Widowed | 5(100.00) | 0(0.00) | | | |
| Separated | 19(90.48) | 2(9.52) | | | |
| Never married | 72(98.63) | 1(1.37) | | | |
| Education (highest completed) | | | 2.06 | 1 | 0.357 |
| Tertiary | 110(91.67) | 10(8.33) | | | |
| Secondary | 158(95.76) | 7(4.24) | | | |
| Primary | 107(93.86) | 7(6.14) | | | |
| Religion* | | | | | |
| Roman Catholic | 69(94.52) | 4(5.48) | | | |
| Protestant | 192(93.20) | 14(6.80) | | 4 | 0.341 |
| Pentecostal | 51(92.73) | 4(7.23) | | | |
| Muslim | 5(83.33) | 1(16.67) | | | |
| Others | 58(98.31) | 1(1.69) | | | |
| Parity | | | | | |
| One child | 135(95.74) | 6(4.26) | | | |
| Two children | 92(94.85) | 5(5.15) | 3.15 | 3 | 0.369 |
| Three children | 72(90.00) | 8(10.00) | | | |
| Four children and above | 76(93.83) | 5(6.17) | | | |
| Age group* | | | | | |
| 11-24 | 169(96.57) | 7(3.43) | | | |
| 25-34 | 159(90.34) | 17(9.66) | | 2 | 0.023 |
| ≥35 | 47(97.52) | 1(2.48) | | | |
| Previous Child with sickle cell disease* | | | | | |
| Yes | 2(100.00) | 0(0.00) | | 1 | 1.000 |
| No | 373(93.95) | 24(6.05) | | | |
| Occupation* | | | | | |
| Student | 70(97.22) | 2(2.78) | | | |
| Formal employment | 34(85.00) | 6(15.00) | | 3 | 0.026 |
| Informal employment | 211(93.36) | 15(6.64) | | | |
| Unemployed | 60(98.36) | 1(1.64) | | | |
| Previous Death of child under 2 years* | | | | | |
| Yes | 28(96.55) | 1(3.45) | | 1 | 0.546 |
| No | 347(93.78) | 23(6.22) | | | |

*Fisher's exact method used where cell contain value <5, p-value<0.05.

support was found to be a strong modifying factor with positive effect (coef = 7.50; std error = 1.31; 95%CI = 4.94–10.07; $p < 0.001$). In the multivariate model, the coefficient of determination($r^2$) was found to be 63.9% (Table 5).

## Discussion

The acceptability of newborn screening for sickle cell disease in the current study was 94%. Maternal occupation and age were strongly associated with acceptability of newborn screening for sickle cell disease. The likelihood of accepting

**Table 3. Association between acceptability of newborn screening for sickle cell disease and knowledge of the postnatal mothers.**

| Characteristics | NBS YES | NBS NO | p-value |
|---|---|---|---|
| Ever heard about SCD | | | 1.000 |
| Yes | 348(93.80) | 23 (6.20) | |
| No | 27(96.40) | 1(3.60) | |
| Knowledge of SCD status | | | 0.612 |
| Yes | 16(100.00) | 0(0.00) | |
| No | 359(93.70) | 24(6.30) | |
| Ever screened for SCD | | | 1.000 |
| Yes | 8(100.00) | 0(0.00) | |
| No | 367(93.90) | 24(6.10) | |
| SCD is blood disorder | | | 1.000 |
| Yes | 340(93.10) | 22(6.10) | |
| No | 35(94.60) | 2(5.40) | |
| SCD is contagious | | | 1.000 |
| Yes | 6(100.00) | 0(0.00) | |
| No | 369(93.90) | 24(6.10) | |
| Maternal inheritance only | | | 1.000 |
| Yes | 2(100.00) | 0(0.00) | |
| No | 373(94.00) | 24(6.00) | |
| Paternal inheritance only | | | 1.000 |
| Yes | 1(100.00) | 0(0.00) | |
| No | 374(94.00) | 24(6.00) | |
| Heard about SCD screening | | | 0.079 |
| Yes | 129(97.00) | 4(3.00) | |
| No | 246(92.30) | 20(7.70) | |

Fisher's exact method used,p-value <0.05.

newborn screening for sickle cell disease is high among postnatal mothers in Homa bay county, especially among those younger than 25 years and those who were students at the time of the study.

The acceptability of newborn screening for sickle cell disease observed in the current study was relatively high (94%). Comparatively, a similar study conducted at the postnatal wards at Kisumu county referral hospital, Kenya, previously observed newborn screening for sickle cell disease acceptability of 99.4% [15] among postnatal mothers. This informed successful implementation of routine newborn screening for sickle cell disease in Kisumu County that is currently ongoing as a consortium. The two studies are both hospital-based with similar design and thus the comparable acceptability. Studies from Nigeria [3,16] observed 86% acceptability of newborn screening for sickle cell disease which is below the acceptability in this current study but is still comparable. The difference between the current study and that of Nnodu and team in Nigeria was that the target population in this study was postnatal mothers while that in Nigeria was conducted in health institutions and University with diverse target population.

Similar studies conducted in Gabon [17] showed low acceptability rates of about 30% despite satisfactory knowledge among the respondents, demonstrating wide variabilities [14]. The reason for the huge difference in the level of acceptability was that most mothers in Gabon feared blood collection from their newborns [17]. This was not the case in Homa bay. There were obvious variations across socio-economic and demographic characteristics, but in both studies younger single mothers were more receptive.

**Table 4. Association between acceptability of newborn screening for sickle cell disease and perception of the postnatal mothers.**

| Characteristics | NBS YES | NBS NO | p-value |
|---|---|---|---|
| SCD occur due to God's will | | | 1.000 |
| True | 27(96.43) | 1(3.57) | |
| False | 348(93.80) | 23(6.20) | |
| SCD occur due to God's punishment | | | 0.268 |
| True | 4(80.00) | 1(20.00) | |
| False | 371(94.16) | 23(5.84) | |
| SCD occur due to evil spirit | | | 0.268 |
| True | 4(80.00) | 1(20.00) | |
| False | 371(94.16) | 23(5.84) | |
| Best timing for SCD Screening | | | 0.180 |
| Prenatally | 4(100.00) | 0(0.00) | |
| 0-6 weeks | 364(94.30) | 22(5.70)) | |
| > 6 weeks | 7(77.78) | 2(22.22) | |
| NBS for SCD is morally right | | | 0.312 |
| True | 370(94.15) | 23(5.85) | |
| False | 5(83.33) | 1(16.67) | |
| NBS is useful for controlling SCD | | | 1.000 |
| True | 360(93.75) | 24(6.25) | |
| False | 15(100.00) | 0(0.00) | |

Fisher's exact method used, p-value <0.05.

**Table 5. Bivariate and Multivariate logistic regression analysis model of acceptability of newborn screening for sickle cell disease.**

| Characteristic | Unadjusted model | | | Adjusted model | | |
|---|---|---|---|---|---|---|
| | OR | 95%CI | P-value | aOR | 95%CI | P-value |
| Occupation | | | | | | |
| Student | REF | | | REF | | |
| Formal employment | 0.16 | 0.03-0.84 | 0.031 | 0.04 | 0.00-0.78 | 0.034 |
| Informal employment | 0.40 | 0.09-1.80 | 0.234 | 0.40 | 0.03-5.74 | 0.504 |
| No employment | 1.71 | 0.15-19.38 | 0.664 | 0.77 | 0.02-24.74 | 0.883 |
| Age category | | | | | | |
| Below 24 | REF | | | REF | | |
| 24-34 years | 0.33 | 0.13-0.86 | 0.024 | 2.30 | 0.38-13.99 | 0.367 |
| ≥35 years | 1.60 | 0.19-13.76 | 0.668 | 1.46 | 0.12-17.17 | 0.720 |
| Cultural support | | | | | | |
| Yes | REF | | | REF | | |
| No | 0.23 | 0.06-0.88 | 0.032 | 4.20 | 0.16-104.18 | 0.381 |
| Male partner support | | | | | | |
| Yes | REF | | | REF | | |
| No | 0.00 | 0.00-0.01 | <0.001 | 0.00 | 0.00-0.01 | <0.001 |

Table legend: model variable significant at 95% CI; p= 0.05; $R^2$= 63.9%.

Maternal occupation was strongly associated with acceptability of newborn screening for sickle cell disease in this study. Those in formal employment were less likely to accept newborn screening for sickle cell disease when compared to the students. This is comparable to the findings of Nnodu et al. in a multicenter survey in Nigeria [3]. The similarity in both studies was that both had a cross-sectional design. The study in Nigeria however had a diverse setting, both in health institutions and universities as opposed to this current study which had a setting only in the health institution. This provides a diverse context to confirm the finding. None the less, contrasting findings were found by Katamea et al.in DRC [7] and Ahmed et al. in Nigeria [18] where occupation was not associated with acceptability. This current study differed with the DRC study in that this was hospital-based study with only postnatal mothers responding to the questionnaire while that of DRC was a community survey with all the adults in the community having a chance to be a respondent. This could have contributed to the differences in findings since not all those adults interviewed were having children. Either, most students are usually younger and thus they may have been left out of the survey disproportionately for having not achieved the country required age cut off for adulthood. The students in this current study were predominantly younger and younger age has been associated with increased acceptability of newborn screening for sickle cell disease.

Maternal age was found to be a strong predictor of acceptability of newborn screening for sickle cell disease. Mothers who were younger than 25 years were found to be 3 times more likely to accept newborn screening than those older. This finding is comparable to the findings of Mombo et al. in a study done in Gabon where mothers younger than 29 years were more likely to accept newborn screening than those older than 29 years [17]. Similarly, a study by Nnodu et al. in Nigeria found that mothers younger than 21 years were more likely to accept newborn screening for SCD than older ones [3]. Younger women were thought to be less culturally bound and therefore more likely to accept newborn screening for SCD [17]. All the 3 studies above had an aspect of hospital-based interviews thus likely the similarity in finding about the age. All the three studies concur that younger maternal age is strongly associated with increased acceptability of newborn screening for sickle cell disease; they only differ in the cut off of the age group.

The multivariate model in this study had the significant factors explaining up to 63% of the variable relationships. The unexplained 37% is likely due to unexplored factors. Review of previous similar studies conducted [7,17] shows diverse factors may be involved but there is no consensus on how different factors interact to influence or potentially affect maternal acceptability of the newborn screening for SCD. Awareness and knowledge levels as well as availability and access to services in high burden areas have been observed to influence acceptability [3].Despite the high willingness of the service recipient, other factors not explored in this study such as cost barriers, service availability and provider factors remain important factors in the overall evaluation of the intervention acceptability.

The study limitations included sampling from one health facility within the county thus the mothers coming from near the health facility could have contributed disproportionately in numbers. However, conducting the study at the county referral hospital was to allow for a representation from all other 8 sub counties within the county with each subcounty contributing at least 5% of the total sample. The assessment of acceptability in a unidimensional approach as opposed to the multidimensional approach was another limitation. However, this was done to allow the comparison with other similar studies.

Primary interventions such as newborn screening program are highly recommended to avert disease complications and early deaths. Acceptance of health technologies is necessary for progressive adoption and continued use by consumers as well as for its implementation at scale [15]. Despite the relatively high acceptability rate especially among the younger mothers, which might imply higher willingness for newborn screening by the recipients, recent studies in Africa show that acceptance of screening services such as newborn screening is diverse and variable over time across different population groups as well as the implementation contexts [7,19]. This might be accounted for by the fact that health technology acceptance is a dynamic concept, and evolves sequentially through a continuum of stages including, pre-use acceptability; initial use acceptability and sustained use acceptability. However, few studies exist on the pre-use health technology acceptance in Africa [8]. The current study examined pre-use acceptability. Whether high acceptability across these stages necessarily translates to scalability remains a multidimensional challenge.

It is,however, important to develop a policy in order to make the intervention universally available and accessible in the county [20]. Such a policy is currently not in place in the county even though health is a devolved function. This is despite the national government putting in place a policy in 2020 recommending routine newborn screening for sickle cell disease. Acceptability of such program had not been evaluated in the county prior. Further, despite the high acceptance and good knowledge on screening for sickle cell disease among the postnatal mothers who were interviewed, only few of the mothers had been screened for sickle cell disease. This was mainly due to the fact that routine screening for sickle cell disease has not been adopted in the county, and including among newborns, and thus only those with symptoms of sickle cell disease are tested or screened for the disease.

Awareness about the intervention or the existence of the intervention helps improve acceptability since those with low knowledge or awareness are less likely to accept the interventions. It will be important that prior to implementation of the newborn screening program in the county, adequate awareness creation is created, and cost barriers eliminated. This has improved acceptability in the HIV care set up [21] and can be extrapolated to sickle cell disease care. Since peak reproductive age is about 24 years, a predominantly young age. Since younger age of the mother has been strongly associated with high acceptability of newborn screening for sickle cell disease, it is expected that the county can easily implement this program especially after addressing access issues.

Implementation of routine newborn screening program for sickle cell disease in all health care facilities in the county with maternity wings is highly recommended in view of the good user acceptability and high prevalence of sickle cell disease at birth in the county. Additionally, there is need to conduct a further study to explore other determinants of acceptability and how they impact on acceptability and actual uptake.

## Conclusion

The acceptability of newborn screening for sickle cell disease among the postnatal mothers in Homa bay county was high, mainly among younger mothers. This high acceptability is indicative of a willingness for screening for sickle cell disease and is necessary for scaling up the program in the general population.

## Supporting information

**S1 Data. Acceptability dataset-jo12Jul2025.**
(CSV)

## Acknowledgments

We acknowledge the postnatal mothers who consented to take part in the study which may improve the policy on newborn screening and the care of patients with sickle cell disease and the research assistants who helped with data collection.

## Author contributions

**Conceptualization:** John Orimbo, Dickens Omondi.

**Data curation:** John Orimbo, Shehu Shagari Awandu, Patrick Owili, Dickens Omondi.

**Formal analysis:** John Orimbo, Shehu Shagari Awandu, Patrick Owili, Dickens Omondi.

**Funding acquisition:** John Orimbo.

**Investigation:** John Orimbo.

**Methodology:** John Orimbo, Shehu Shagari Awandu, Faith Muhonja, Patrick Owili, Dickens Omondi.

**Project administration:** John Orimbo.

**Resources:** John Orimbo, Dickens Omondi.

**Software:** John Orimbo.

**Supervision:** Shehu Shagari Awandu, Faith Muhonja, Patrick Owili, Dickens Omondi.

**Validation:** John Orimbo, Shehu Shagari Awandu, Faith Muhonja, Patrick Owili, Dickens Omondi.

**Visualization:** John Orimbo, Dickens Omondi.

**Writing – original draft:** John Orimbo.

**Writing – review & editing:** John Orimbo, Shehu Shagari Awandu, Faith Muhonja, Patrick Owili, Dickens Omondi.

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
