## [Decision Letter · Decision Letter 0]

Dear Dr. Orimbo,

Thank you for submitting your manuscript to PLOS ONE. After careful consideration, we feel that it has merit but does not fully meet PLOS ONE’s publication criteria as it currently stands. Therefore, we invite you to submit a revised version of the manuscript that addresses the points raised during the review process.

We look forward to receiving your revised manuscript.

Kind regards,

Ibrahim Sebutu Bello, MBBS, MPH, MD, FMCGP

Academic Editor

PLOS ONE

Journal Requirements:

3. Thank you for stating the following in your Competing Interests section: None

Reviewers' comments:

Reviewer's Responses to Questions

**Comments to the Author**

1. Is the manuscript technically sound, and do the data support the conclusions?

Reviewer #1: Yes

Reviewer #2: Yes

2. Has the statistical analysis been performed appropriately and rigorously?

Reviewer #1: Yes

Reviewer #2: Yes

3. Have the authors made all data underlying the findings in their manuscript fully available?

Reviewer #1: No

Reviewer #2: No

4. Is the manuscript presented in an intelligible fashion and written in standard English?

Reviewer #1: Yes

Reviewer #2: Yes

Reviewer #1: well written manuscript. Some comments: -

1. Introduction- Could you highlight the national government newborn screening policy on SCD? This is referenced in the methods and discussion section.

2. Its likely the data are clustered around the study facility. Its important to provide the geographical location of the participants - the data shown in the discussion under limitations 'with each subcounty contributing at least 5% of the

369 total sample has not been provided.

3. Consider dropping this phrase 'at Homabay County Teaching and Referral Hospital, western Kenya' to improve readability

4. Consider dropping figure 1 as it does not add much value

5. What informed the questions on acceptability? They seem narrow focusing on beliefs. There might be other key aspects of acceptability that these narrowly focused questions would not evaluate. its also not clear how responses to these questions could inform improved acceptability in other areas.

6. Screening Barriers: Could you comment on why, despite high acceptance and knowledge of SCD screening, the majority of the mothers reported not having been screened? What could be the barriers to screening that would need specific recommendations?

7. was the effect of clustering in time and space explored in this study

8 The discussion needs to be revised to be succinct and clear-- a maximum of two pages

9. Read the paper to correct grammar and appropriate punctuations

Reviewer #2: Summary

The authors assessed acceptability of newborn screening for Sickle Cell disease among 399 post-natal mothers at Homa Bay county hospital in western Kenya. They focused on assessing level of acceptability as a step in the pathway towards programmatic scale up of newborn screening.

Mothers who chose to participate were consented and interviewed using a structured questionnaire and their newborns were screened using Sickle SCAN point-of-care test.

Acceptability of screening was high at 94%. Being a student was associated with a higher likelihood of accepting screening. The authors recommended that Homabay county and the Kenyan ministry of health should implement routine newborn screening for sickle cell disease in all level 2-6 hospitals.

General Comment

The manuscript is clearly written and addresses an important topic.

It could however benefit from some edits. Authors should pay particular attention to data analysis and interpretation of results on factors associated with acceptability screening.

Abstract

1. Methods (Line 21-27): It is not clear whether potential participants were empowered with information on the benefits of screening newborns for SCD before being invited to make an informed choice to enroll in the study.

2. Conclusion (Line 35-36): The second sentence is directive and not fully supported by the results. Why all level 2-6 hospitals? Consider editing this sentence to improve its clarity.

Background

It is stated that the prevalence of sickle cell disease and sickle cell trait in western Kenya is 4.5% and 18%, respectively. Please specify age band for reported estimates.

Methods

It is stated that Minor mothers also were provided with a signed assent… edit this to improve clarity. Specify what was done. Assented and signed an assent form or assented verbally and someone else signed a consent form on their behalf?

There are no details of how mothers with no prior knowledge on sickle disease were empowered with information to make informed choice about participating in the study is missing. Consider editing to include the missing information.

Line 127: To is misplaced within the sentence: Consider deleting.

Results

Line 191: Replace “belived” with believed.

The confidence intervals for some estimates are uncomfortably wide thus undermining the utility of some results. Please review. E.g. line 32-33: being a student (aOR= 25.02; 95% CI=1.29-484.51; p= 0.033).

Table 5:

1. Review choice of reference group for population categories in table 5.

2. The authors have only used 4 variables to check associations based on chi square tabulations instead of using all the variables in the unadjusted model to determine those to be used in adjusted model.

3. Overall, the very wide confidence intervals may undermine the practical applications of results touching on the predictors acceptability.

**Do you want your identity to be public for this peer review?** For information about this choice, including consent withdrawal, please see our Privacy Policy

Reviewer #1: **Yes: ** Peninah Munyua

Reviewer #2: No

---

## [Author Response · Author response to Decision Letter 1]

11 Feb 2025

1. Introduction- Could you highlight the national government newborn screening policy on SCD? This is referenced in the methods and discussion section.

Response: This has been added to the abstract to read, “The Kenyan ministry of health, in 2020, published a policy on newborn screening for sickle cell disease from levels 2 to 6 healthcare facilities.”

2. Its likely the data are clustered around the study facility. Its important to provide the geographical location of the participants - the data shown in the discussion under limitations 'with each subcounty contributing at least 5% of the

369 total sample has not been provided.

Response: This has now been provided in the table 1 of the results section under the sociodemographic characteristics of the participants

3. Consider dropping this phrase 'at Homabay County Teaching and Referral Hospital, western Kenya' to improve readability

Response: This has been dropped in all other parts except the abstract where it is retained to explain the study setting

4. Consider dropping figure 1 as it does not add much value

Response: Figure one has been dropped

5. What informed the questions on acceptability? They seem narrow focusing on beliefs. There might be other key aspects of acceptability that these narrowly focused questions would not evaluate. its also not clear how responses to these questions could inform improved acceptability in other areas.

Response: The questions were largely informed by the review of literature on the previous studies on the factors that have influenced user acceptability especially at user level and guided by the theoretical framework of acceptability. Other factors like access and provider attitude which affect majorly concurrent acceptability were not explored in this study since newborn screening is not yet implemented in the county.

6. Screening Barriers: Could you comment on why, despite high acceptance and knowledge of SCD screening, the majority of the mothers reported not having been screened? What could be the barriers to screening that would need specific recommendations?

Response: This has been added to the discussion section on paragraph 3 to read, “Further, despite the high acceptance and good knowledge on screening for sickle cell disease among the postnatal mothers who were interviewed, only few of the mothers had been screened for sickle cell disease. This was mainly due to the fact that routine screening for sickle cell disease has not been adopted in the county, and including among newborns, and thus only those with symptoms of sickle cell disease are tested or screened for the disease.”

7. was the effect of clustering in time and space explored in this study

Response: No, this was not explored

8 The discussion needs to be revised to be succinct and clear-- a maximum of two pages

Response: This has been revised

9. Read the paper to correct grammar and appropriate punctuations

Response: Grammar and punctuations have been corrected across the paper.

Reviewer #2: Summary

The authors assessed acceptability of newborn screening for Sickle Cell disease among 399 post-natal mothers at Homa Bay county hospital in western Kenya. They focused on assessing level of acceptability as a step in the pathway towards programmatic scale up of newborn screening.

Mothers who chose to participate were consented and interviewed using a structured questionnaire and their newborns were screened using Sickle SCAN point-of-care test.

Acceptability of screening was high at 94%. Being a student was associated with a higher likelihood of accepting screening. The authors recommended that Homabay county and the Kenyan ministry of health should implement routine newborn screening for sickle cell disease in all level 2-6 hospitals.

General Comment

The manuscript is clearly written and addresses an important topic.

It could however benefit from some edits. Authors should pay particular attention to data analysis and interpretation of results on factors associated with acceptability screening.

Abstract

1. Methods (Line 21-27): It is not clear whether potential participants were empowered with information on the benefits of screening newborns for SCD before being invited to make an informed choice to enroll in the study.

Response: All the potential participants were taken through written informed consent prior to participation. The informed consent had both the risks and benefits of participation in the study, including of doing screening for sickle cell disease on the newborn, and this were adequately explained to the participant prior. Only participants who were willing after being taken through the informed consent signed the consent and were subsequently enrolled. This thus met the threshold of adequate informed to allow them make an informed decision. This has been added to the abstract to read, “After obtaining informed consent, a semi-structured questionnaire was used for data collection.”

2. Conclusion (Line 35-36): The second sentence is directive and not fully supported by the results. Why all level 2-6 hospitals? Consider editing this sentence to improve its clarity.

Response: This has been revised to read, “The Homabay county ministry of health should implement routine newborn screening for sickle cell disease in all health care facilities conducting deliveries of newborns”. This is since the study was conducted among post delivery mothers, it can be extrapolated for all healthcare facilities conducting deliveries across the county.

Background

It is stated that the prevalence of sickle cell disease and sickle cell trait in western Kenya is 4.5% and 18%, respectively. Please specify age band for reported estimates.

Response: This statement has been revised to read, “The prevalence of sickle cell disease around western Kenya at birth is about 4.5% for sickle cell disease, and 18% for sickle cell trait”

Methods

It is stated that Minor mothers also were provided with a signed assent… edit this to improve clarity. Specify what was done. Assented and signed an assent form or assented verbally and someone else signed a consent form on their behalf?

Response: This has been edited to read, “Written informed consent was obtained from potential participants who were 18years and above. A written informed assent was obtained from minor mothers (those less than 18 years and lacking legal capacity to consent on their own) in addition to a written consent from their parents or guardians.”

There are no details of how mothers with no prior knowledge on sickle disease were empowered with information to make informed choice about participating in the study is missing. Consider editing to include the missing information.

Response: This information was provided to the mothers through the informed consent and assent and has now been added to read, “Written informed consent was obtained from potential participants who were 18years and above. A written informed assent was obtained from minor mothers (those less than 18 years and lacking legal capacity to consent on their own) in addition to a written consent from their parents or guardians. The informed consents and assents were adequate details on sickle cell disease and their participation in the study, and also on risk and benefits of participation. This was made so in order to provide the potential participants with adequate prior information for making an informed decision before signing the informed consent or assent.”

Line 127: To is misplaced within the sentence: Consider deleting.

Response: This has been deleted

Results

Line 191: Replace “belived” with believed.

Response: The spelling has been corrected to believed.

The confidence intervals for some estimates are uncomfortably wide thus undermining the utility of some results. Please review. E.g. line 32-33: being a student (aOR= 25.02; 95% CI=1.29-484.51; p= 0.033).

Response: This has been reviewed

Table 5:

1. Review choice of reference group for population categories in table 5.

Response: This has been reviewed

2. The authors have only used 4 variables to check associations based on chi square tabulations instead of using all the variables in the unadjusted model to determine those to be used in adjusted model.

Response: Only the variables that had a statistically significant association with the acceptability of newborn screening for sickle cell disease were moved to unadjusted model to determine the strength of association thus only 4 variables moved.

3. Overall, the very wide confidence intervals may undermine the practical applications of results touching on the predictors acceptability.

Response: This has been reviewed upon review of the choice of reference group

---

## [Decision Letter · Decision Letter 1]

Dear Dr. Orimbo,

Overall, it is a well written manuscript.

Kindly consider a minor revision of the manuscript based on the following comments:

    Introduction - Are there no studies in literature on acceptability of newborn screening at all?  You may please consider and review articles such as the article by “Nnachi OC et. Acceptability of Newborn Screening for Sickle Cell Disease among Post-Partum Mothers in Abakaliki, South East Nigeria. West Afr J Med. 2023 Mar 31;40(3):298-304. PMID: 37017939.” And other related literature.

    Methods – Line 117 - Did you obtain assent from children? Assent is not the same as consent. Your manuscript implies they are similar. Please clarify what you did. Assent is usually obtained from minors, and consent from adults.

    Methods – Line 131 – 133. There’s need to describe the government policy in a section in the initially under methods.

    Methods – Linie 181 - Assent is not the same as consent.

    Discussion – Line 314 – 425.

Please revise the discussion section. Kindly write this section in a systematic, focused manner. i. Start with a Summary of Key Findings (May be 2-3 key findings). ii. Interpret the Findings. iii. Compare with Existing Literature. Iv. Explain the Strengths and Limitations. v. Discuss Implications for Practice or Policy. vi. Offer Recommendations or Future Research Directions. vii.Write a clear conclusion.

While most elements of the discussion are present in the Discussion Section in its present form, the discussion section will benefit from a re-organisation.

In addition, the authors are also advised to address the peer reviewer's comments below:

"Summary

The authors assessed acceptability of newborn screening for Sickle Cell disease among 399 post-natal mothers at Homa Bay county hospital in western Kenya. They focused on assessing level of acceptability as a step in the pathway towards programmatic scale up of newborn screening.

Mothers who chose to participate were consented and interviewed using a structured questionnaire and their newborns were screened using Sickle SCAN point-of-care test.

Acceptability of screening was high at 94%. Being a student was associated with a higher likelihood of accepting screening. The authors recommended that Homabay county and the Kenyan ministry of health should implement routine newborn screening for sickle cell disease in all level 2-6 hospitals.

General Comment

The manuscript is clearly written and addresses an important topic.

It could however benefit from some edits. Authors should pay particular attention to data analysis and interpretation of results on factors associated with acceptability screening.

Abstract

1. Methods (Line 21-27): It is not clear whether potential participants were empowered with information on the benefits of screening newborns for SCD before being invited to make an informed choice to enroll in the study.

2. Conclusion (Line 35-36): The second sentence is directive and not fully supported by the results. Why all level 2-6 hospitals? Consider editing this sentence to improve its clarity.

Background

It is stated that the prevalence of sickle cell disease and sickle cell trait in western Kenya is 4.5% and 18%, respectively. Please specify age band for reported estimates.

Methods

It is stated that Minor mothers also were provided with a signed assent… edit this to improve clarity. Specify what was done. Assented and signed an assent form or assented verbally and someone else signed a consent form on their behalf?

There are no details of how mothers with no prior knowledge on sickle disease were empowered with information to make informed choice about participating in the study is missing. Consider editing to include the missing information.

Line 127: To is misplaced within the sentence: Consider deleting.

Results

Line 191: Replace “belived” with believed.

The confidence intervals for some estimates are uncomfortably wide thus undermining the utility of some results. Please review. E.g. line 32-33: being a student (aOR= 25.02; 95% CI=1.29-484.51; p= 0.033).

Table 5:

1. Review choice of reference group for population categories in table 5.

2. The authors have only used 4 variables to check associations based on chi square tabulations instead of using all the variables in the unadjusted model to determine those to be used in adjusted model.

3. Overall, the very wide confidence intervals may undermine the practical applications of results touching on the predictors acceptability."

We look forward to receiving your revised manuscript.

Kind regards,

Abiola Olukayode Olaleye, MBBS, MPH, FWACP

Academic Editor

PLOS ONE

Journal Requirements:

Reviewers' comments:

Reviewer's Responses to Questions

**Comments to the Author**

Reviewer #2: All comments have been addressed

2. Is the manuscript technically sound, and do the data support the conclusions?

Reviewer #2: Yes

3. Has the statistical analysis been performed appropriately and rigorously?

Reviewer #2: Yes

4. Have the authors made all data underlying the findings in their manuscript fully available?

Reviewer #2: Yes

5. Is the manuscript presented in an intelligible fashion and written in standard English?

Reviewer #2: Yes

Reviewer #2: (No Response)

**Do you want your identity to be public for this peer review?** For information about this choice, including consent withdrawal, please see our Privacy Policy

Reviewer #2: No

---

## [Author Response · Author response to Decision Letter 2]

11 Jun 2025

Introduction - Are there no studies in literature on acceptability of newborn screening at all? You may please consider and review articles such as the article by “Nnachi OC et. Acceptability of Newborn Screening for Sickle Cell Disease among Post-Partum Mothers in Abakaliki, South East Nigeria. West Afr J Med. 2023 Mar 31;40(3):298-304. PMID: 37017939.” And other related literature.

Response: Thank you for the input. However, the introduction reads. “There is however, paucity of data on the acceptability of newborn screening in the county” This implies there has not been any study previous assessing acceptability in Homa bay county. We however acknowledge that several studies have been done across sub-Saharan Africa on acceptability of newborn screening for sickle cell disease.

Methods – Line 117 - Did you obtain assent from children? Assent is not the same as consent. Your manuscript implies they are similar. Please clarify what you did. Assent is usually obtained from minors, and consent from adults.

Response: Consent was obtained from adult mothers. For minor mothers, they signed an assent and their parent or guardian signed the consent. This has now been clarified.

Methods – Line 131 – 133. There’s need to describe the government policy in a section in the initially under methods.

Response: Thank you. The policy has been explained under the introduction section.

Methods – Linie 181 - Assent is not the same as consent.

Response: Thank you, this has been clarified.

Discussion – Line 314 – 425.

Please revise the discussion section. Kindly write this section in a systematic, focused manner. i. Start with a Summary of Key Findings (May be 2-3 key findings). ii. Interpret the Findings. iii. Compare with Existing Literature. Iv. Explain the Strengths and Limitations. v. Discuss Implications for Practice or Policy. vi. Offer Recommendations or Future Research Directions. vii.Write a clear conclusion.

While most elements of the discussion are present in the Discussion Section in its present form, the discussion section will benefit from a re-organisation.

Response: This has been revised appropriately as guided.

In addition, the authors are also advised to address the peer reviewer's comments below:

"Summary

The authors assessed acceptability of newborn screening for Sickle Cell disease among 399 post-natal mothers at Homa Bay county hospital in western Kenya. They focused on assessing level of acceptability as a step in the pathway towards programmatic scale up of newborn screening.

Mothers who chose to participate were consented and interviewed using a structured questionnaire and their newborns were screened using Sickle SCAN point-of-care test.

Acceptability of screening was high at 94%. Being a student was associated with a higher likelihood of accepting screening. The authors recommended that Homabay county and the Kenyan ministry of health should implement routine newborn screening for sickle cell disease in all level 2-6 hospitals.

General Comment

The manuscript is clearly written and addresses an important topic.

It could however benefit from some edits. Authors should pay particular attention to data analysis and interpretation of results on factors associated with acceptability screening.

Abstract

1. Methods (Line 21-27): It is not clear whether potential participants were empowered with information on the benefits of screening newborns for SCD before being invited to make an informed choice to enroll in the study.

Response: The potential participants were provided with the information through the informed consent documents which they were taken through by a research assistant in a language of their choice before making informed decision to enrol. The information included benefits of screening the newborn, result interpretation, linkages to care and possible risks of participation in the study.

2. Conclusion (Line 35-36): The second sentence is directive and not fully supported by the results. Why all level 2-6 hospitals? Consider editing this sentence to improve its clarity.

Response: This has been edited to read, “The acceptability of newborn screening for sickle cell disease is high in the county. The Homabay county ministry of health should implement routine newborn screening for sickle cell disease in all healthcare facilities conducting deliveries of newborns.”

Background

It is stated that the prevalence of sickle cell disease and sickle cell trait in western Kenya is 4.5% and 18%, respectively. Please specify age band for reported estimates.

Response: This has been clarified to read, “The prevalence of sickle cell disease around western Kenya at birth is about 4.5% for sickle cell disease, and 18% for sickle cell trait.”

Methods

It is stated that Minor mothers also were provided with a signed assent… edit this to improve clarity. Specify what was done. Assented and signed an assent form or assented verbally and someone else signed a consent form on their behalf?

There are no details of how mothers with no prior knowledge on sickle disease were empowered with information to make informed choice about participating in the study is missing. Consider editing to include the missing information.

Response: This has been added in the methods section under the sub topic enrolment and data collection, lines 184-193

Line 127: To is misplaced within the sentence: Consider deleting.

Response: This was deleted.

Results

Line 191: Replace “belived” with believed.

Response: This was updated accordingly

The confidence intervals for some estimates are uncomfortably wide thus undermining the utility of some results. Please review. E.g. line 32-33: being a student (aOR= 25.02; 95% CI=1.29-484.51; p= 0.033).

Response: This was reviewed to “(aOR= 0.04; 95% CI=0.00-0.78; p= 0.033)” after re-analysis

Table 5:

1. Review choice of reference group for population categories in table 5.

Reviewed: This has been reviewed.

2. The authors have only used 4 variables to check associations based on chi square tabulations instead of using all the variables in the unadjusted model to determine those to be used in adjusted model.

Response: All the variables were subject ton test of association using either chi-square or fisher’s exact, where criteria for chi-square was not met. Only the variables that had an association with acceptability of newborn screening for sickle cell disease were subjected with univariate logistic regression (unadjusted model) in order to determine the strength of association thus not all variables were used for the unadjusted model. This was as per the approved protocol statistical analysis plan.

3. Overall, the very wide confidence intervals may undermine the practical applications of results touching on the predictors acceptability."

Response: upon re-analysis, the confidence intervals are now much better. However, this can be attributed to slightly smaller sample size even though our sample size was larger than the calculated sample size.

---

## [Editor Report · Decision Letter 2]

High acceptability of newborn screening for sickle cell disease among post-natal mothers in western Kenya

PLOS ONE

Dear Dr.  Orimbo,

Thank you for submitting your manuscript to PLOS ONE. After careful consideration, we feel that it has merit but does not fully meet PLOS ONE’s publication criteria as it currently stands. Therefore, we invite you to submit a revised version of the manuscript that addresses the points raised during the review process.

**Please revise the title of the manuscript as follows and resubmit:**

**
*"High Acceptability of Newborn Screening for Sickle Cell Disease Among Post-Natal Mothers in Western Kenya"*
**

We look forward to receiving your revised manuscript.

Kind regards,

Abiola Olukayode Olaleye, MBBS, MPH, FWACP

Academic Editor

PLOS ONE
---

## [Author Response · Author response to Decision Letter 3]

4 Jul 2025

The title has been updated to read, "High Acceptability of Newborn Screening for Sickle Cell Disease Among Post-Natal Mothers in Western Kenya "as was suggested

---

## [Editor Report · Decision Letter 3]

High acceptability of newborn screening for sickle cell disease among post-natal mothers in western Kenya

PONE-D-24-20006R3

Dear Dr. Orimbo,

We’re pleased to inform you that your manuscript has been judged scientifically suitable for publication and will be formally accepted for publication once it meets all outstanding technical requirements.

Kind regards,

Abiola Olukayode Olaleye, MBBS, MPH, FWACP

Academic Editor

PLOS ONE

---

## [Editor Report · Acceptance letter]

PONE-D-24-20006R3

PLOS ONE

Dear Dr. Orimbo,

I'm pleased to inform you that your manuscript has been deemed suitable for publication in PLOS ONE. Congratulations! Your manuscript is now being handed over to our production team.

Kind regards,

on behalf of

Dr. Abiola Olukayode Olaleye

Academic Editor

PLOS ONE